# MEMPHIS: a smartphone app using psychological approaches for women with chronic pelvic pain presenting to gynaecology clinics: a randomised feasibility trial

Gordon Forbes,[1] Sian Newton,[2] Clara Cantalapiedra Calvete,[3] Judy Birch,[4] Julie Dodds,[5] Liz Steed,[2] Carol Rivas  ,[6] Khalid Khan,[7] Frank Röhricht,[8] Stephanie Taylor,[9] Brennan C Kahan,[10] Elizabeth Ball[3]

► https://doi.org/10.1136/bmjopen-2019-030711

For numbered affiliations see end of article.

**Correspondence to**
Dr Elizabeth Ball;
elizabeth.ball9@nhs.net

## ABSTRACT

**Objectives** To evaluate the feasibility of a randomised trial of a modified, pre-existing, mindfulness meditation smartphone app for women with chronic pelvic pain.

**Design** Three arm randomised feasibility trial.

**Setting** Women were recruited at two gynaecology clinics in the UK. Interventions were delivered via smartphone or computer at a location of participants choosing.

**Participants** Women were eligible for the study if they were over 18, had been experiencing organic or non-organic chronic pelvic pain for 6 months or more, and had access to a computer or smartphone. 90 women were randomised.

**Interventions** Daily mindfulness meditation delivered by smartphone app, an active control app which delivered muscle relaxation techniques, and usual care without app. Interventions were delivered over 60 days.

**Primary and secondary outcome measures** Outcomes included length of recruitment, follow-up rates, adherence to the app interventions, and clinical outcomes measured at baseline, two, three and 6 months.

**Results** The target sample size was recruited in 145 days. Adherence to the app interventions was extremely low (mean app use 1.8 days mindfulness meditation group, 7.0 days active control). Fifty-seven (63%) women completed 6-month follow-up, and 75 (83%) women completed at least one postrandomisation follow-up. The 95% CIs for clinical outcomes were consistent with no benefit from the mindfulness meditation app; for example, mean differences in pain acceptance scores at 60 days (higher scores are better) were −2.3 (mindfulness meditation vs usual care, 95% CI: −6.6 to 2.0) and −4.0 (mindfulness meditation vs active control, 95% CI: −8.1 to 0.1).

**Conclusions** Despite high recruitment and adequate follow-up rates, demonstrating feasibility, the extremely low adherence suggests a definitive randomised trial of the mindfulness meditation app used in this study is not warranted. Future research should focus on improving patient engagement.

**Trial registration numbers** NCT02721108; ISRCTN10925965; Results.

### Strengths and limitations of this study

► This is a randomised feasibility study designed specifically to test whether evaluation of the intervention is viable in a full-scale randomised trial.

► The trial achieved target recruitment demonstrating feasibility of recruiting patients to trials of apps for women experiencing chronic pelvic pain.

► Measures of adherence to the app interventions were robust and complete as they relied on system-generated data.

► This trial evaluated only one app provided by a leading developer of mindfulness meditation apps.

## BACKGROUND

Chronic pelvic pain in women is defined as intermittent or constant pain in the lower abdomen or pelvis for 6 or more months, and affects more than 24% of women worldwide.[1] It has considerable impact on patients' quality of life, including their mental health and their income due to loss of working days and diminished work capacity.[2] Chronic pelvic pain may or may not have an identifiable pathology and has both physical and psychological contributors.[3] Chronic pelvic pain is difficult to treat but health outcomes can be improved by psychological and lifestyle interventions.[4 5] However, these are often not received[6 7] due to difficulties in access or service shortages.

Systematic reviews of randomised controlled trials evaluating mindfulness meditation have shown benefit in chronic pain conditions (positive effects on depression, quality of life and pain symptoms).[8 9] Mindfulness is a form of meditation where the client attempts to maintain attention on the present moment, for example by focusing their attention on

their breathing. Whenever attention wanders from the present moment to thoughts and feelings, the client will simply take notice of them and let them go as attention is returned to the present. There is an emphasis on simply taking notice of whatever the mind happens to wander to and accepting each object without making judgements about it or elaborating on its implications additional meaning or need for action. The client is further encouraged to use the same general approach outside of their formal meditation practice, bringing awareness back to the here and now, whenever they notice a general lack of awareness or they notice that attention has become focused on streams of thoughts and worries.[10] So far no randomised controlled trials of mindfulness meditation exist in chronic pelvic pain in women, but results from uncontrolled studies comparing pretreatment and posttreatment outcomes have suggested there may be a benefit (such as increased ability to control pain, improvements in mental health, emotional well-being, work and family life and social functioning).[11 12]

Mindfulness meditation can be resource intensive and typically requires multiple face-to-face visits over a period of weeks or months.[13] If effective, delivery of mindfulness meditation via smartphone app to women with chronic pelvic pain could provide a new treatment option for this patient group, requiring a minimal increase in resources for healthcare systems. No studies have evaluated mindfulness mediation via smartphone app for women with chronic pelvic pain. We therefore conducted a randomised feasibility trial to assess the feasibility of a future full-scale, multicentre randomised trial of a mindfulness meditation intervention delivered by the Headspace smartphone app (Headspace Ltd) for patients with chronic pelvic pain.

The primary objective of the study was to assess the feasibility of implementing a randomised trial of a mindfulness meditation intervention delivered by a smartphone app for women with chronic pelvic pain. Specifically, we assessed feasibility of recruitment, levels of adherence to the intervention, and estimated parameters required for the sample size calculation for a full trial. Secondary objectives were to measure the clinical outcomes that may be used in a future full-scale trial and make estimates for the effect of the intervention. We examined a variety of clinical outcomes assessing pain acceptance and self-efficacy, pain-related disability, mental health, mindfulness, and sexual health, and quality of life. No primary outcome was specified because this was a feasibility study;[14] however, it was anticipated that chronic pain acceptance would be the primary outcome for any future study assessing effectiveness. Pain acceptance was chosen by the study group with input from pain patients and clinicians because it has been shown to be a meaningful clinical outcome that was improved by mindfulness mediation in other pain conditions.[8] This article reports quantitative findings; qualitative findings will be published separately.[15]

## METHODS

### Study design and participants

This three-arm parallel group randomised feasibility trial was conducted at two gynaecology clinics within Barts Health National Health System (NHS) trust. Eligible patients were aged 18 years or over had been experiencing chronic pelvic pain with or without identifiable pathology (ie, organic or non-organic chronic pelvic pain) for 6 months or more, and understood simple English. Patients were excluded from the trial if they had no access to a personal computer or smartphone, or were current users of the publicly available Headspace app. Patients were recruited via pelvic pain or endometriosis clinics at participating sites as well as at other routine appointments. Prior to randomisation, all participants were provided with a patient information sheet and provided written informed consent. The study protocol has been published[16] and the final version is given in online supplementary appendix 1.

### Interventions

Full details of the interventions are available in the published protocol.[16] Patients were randomised to receive mindfulness meditation, an active control or usual care only. All participants received usual care, which included watch and wait, medication and/or surgery.

Women in the mindfulness meditation group received access to a 60-day progressive mindfulness meditation course delivered via the Headspace app. The intervention consisted of daily, audio-guided, mindfulness meditation sessions. The first 10 days of the course taught basics of mindfulness meditation. Following this, participants were able to access the module on meditation that was targeted at chronic pain. This module was specifically designed for the MEMPHIS trial. Session length was 10 min for the first 10 days, 15 min up to day 20 and 20 min up to day 60.

The active control group received access to a series of muscle relaxation sessions. These sessions were identical every day, except that their duration increased to mirror the increasing duration of the meditation content being listened to by the intervention group.

Women in the mindfulness mediation group and active control group were given instructions on how to install the app. No further face-to-face induction was given on how to carry out the techniques taught in the apps. To maintain blinding between the mindfulness meditation group and active control, both groups accessed their intervention via the same app, and received instructions for the same duration, delivered by the same narrator. Only the content of the instructions differed.

We chose to evaluate an existing commercial app teaching mindfulness by guided meditation (Headspace Ltd) as this approach was expected to save time and money compared with designing a new app from scratch. The Headspace app was adapted for use by chronic pelvic pain patients by augmenting the existing app with a novel module on chronic pain, which could be accessed

after completing 10 days of basic training in mindfulness meditation.

## Randomisation and blinding

Women were randomly allocated 1:1:1 to the active intervention app, active control app or treatment as usual using random-permuted blocks (block size 27, 30, 33) without stratification using a centralised web-based service with allocation concealment. The randomisation list was generated using the Pragmatic Clinical Trials Unit's randomisation system using a random number generator. Following randomisation, participants, recruiting staff and researchers conducting follow-up interviews were not blinded to whether allocation was to the treatment as usual group or to one of the app groups (mindfulness meditation or active control); however, for allocation to an app group they were blinded to which specific app group this was (mindfulness meditation or active control). The trial statisticians remained blinded to allocation until the statistical analysis plan had been signed off, all data collection was completed, and the dataset was finalised.

## Data collection

Data on patient adherence to the app were collected by Headspace Ltd. Data collection was performed automatically by the app and recorded every time a participant completed more than 90% of a session with the app. No data were collected on sessions that were less than 90% complete. Headspace provided the trial team with a list of codes, which were linked to the randomisation system, and given to trial participants to access the app. At the end of the trial, data on completed sessions were transferred via a secure file transfer protocol from Headspace to the trial team. No data that could identify participants were included in this transfer. Clinical outcome measures were collected in person at baseline prior to randomisation and via postal questionnaires or telephone at 2, 3 and 6 months post randomisation. App satisfaction and usability questionnaires were collected via postal questionnaires or telephone. Shopping vouchers (£5), text reminders and phone calls were introduced to improve follow-up rates 3 months after recruitment began: shopping vouchers were sent by post with each follow-up questionnaire; participants were sent text reminders and up to three attempts were made to contact participants by phone if questionnaire responses were not received within 10 days.

## Outcomes

Feasibility outcomes were: time to recruit 90 patients to the study; SD of chronic pain acceptance questionnaire (CPAQ-8)[17] (as this was likely to be the primary outcome for a future full-scale trial); proportion of participants completing a follow-up questionnaire at 6 months post randomisation; and proportion of participants not returning a follow-up questionnaire by post but who answered a telephone questionnaire at 6 months. SD of CPAQ was included as an outcome as this information

would be required for the sample size calculation for a full trial. App usability was measured using the system usability scale[18] and a purpose made, non-validated questionnaire developed from patient and public involvement (PPI) group discussion. Adherence to the app interventions was measured in the following ways:

a. Number of days a patient has used the app within 60 days of randomisation.
b. Number of weeks a patient has used the app on 3 or more days within the first 8 weeks from randomisation.
c. Whether the patient has used the app on at least 22 days within 60 days of randomisation (binary outcome).
d. Whether the patient has used the app on 3 or more days in 6 or more weeks within the first 8 weeks of randomisation (binary outcome).
e. Whether the patient has used the app on 22 or more days within the first 60 days from randomisation and used the app on 3 or more days in 6 or more weeks within the first 8 weeks from randomisation (binary outcome).

Measures of app use were chosen following discussion within the trial management group and trial-steering group to give a complete picture of how participants were using the app. App use was defined as having completed at least 90% of a session. This definition of app use was changed after the trial started recruiting but before any data were analysed due to a change in the way data on app use were collected by Headspace. The original definition of app use was for patients to have completed at least 50% of a session.

The following clinical outcomes were measured at baseline, 60 days, 3 months and 6 months post randomisation:

a. Pain acceptance score (measured by the CPAQ-8).[17]
b. Pain-related disability (chronic pain grade)—disability subscale).[19]
c. Quality-of-life subscales (measured by the RAND short form 36 health survey): social-functioning subscale, pain-functioning subscale and general health subscale.[20]
d. The depression and anxiety subscales of the Hospital Anxiety and Depression Scale).[21]
e. Mindfulness (cognitive and mindfulness—revised scale).[22]
f. Self-efficacy (pain self-efficacy questionnaire).[23]
g. Sexual health among sexually active participants (sexual health outcomes in women questionnaire [SHOW-Q]).[24]
h. Sexual health pelvic problem interference score (SHOW-Q pelvic problem subscale).[24]
i. An individualised outcome (measure yourself medical outcome profile (MYMOP)).[25]

## Statistical analysis

A sample size of 90 participants was chosen as it would provide a precise estimate for the SD of the primary clinical outcome (likely to be pain acceptance),[26 27] which could be used to inform the sample size calculation of a subsequent full-scale trial. This sample size is also

adequate to provide estimates of proportions for binary outcomes.[27]

Feasibility outcomes and baseline data were summarised using descriptive statistics. Clinical outcomes were analysed using a linear mixed-effects models with outcome measurement (at three follow-up time points) as the dependent variable and an unstructured correlation matrix for the residuals.[28] The model included fixed effects for time, treatment arm, time-by-treatment interactions and baseline measure of the outcome.[29] Analysis was by intention-to-treat; all patients with an observed outcome for at least one of the three follow-up time points were included in the analysis,[30] and were analysed according to their randomised group. Missing baseline clinical measures were handled using mean imputation.[31] See online supplementary appendix 2 for a full statistical analysis plan.

### Patient and public involvement

The study design and intervention was discussed with a PPI group formed of 15 women who attended the recruiting clinics. A basic version of the app by Headspace Ltd was made available to the group for testing. A patient, who bought their own experience and acted as a representative for a charity supporting those with chronic pelvic pain (CPP), sat on the trial management group that oversaw the conduct of the trial.

## RESULTS
### Feasibility outcomes
Ninety women were recruited to the trial in 145 days between May 2016 and September 2016. A Consolidated Standards of Reporting Trials diagram is shown in figure 1 and baseline characteristics are shown in table 1, with additional baseline data given in online supplementary

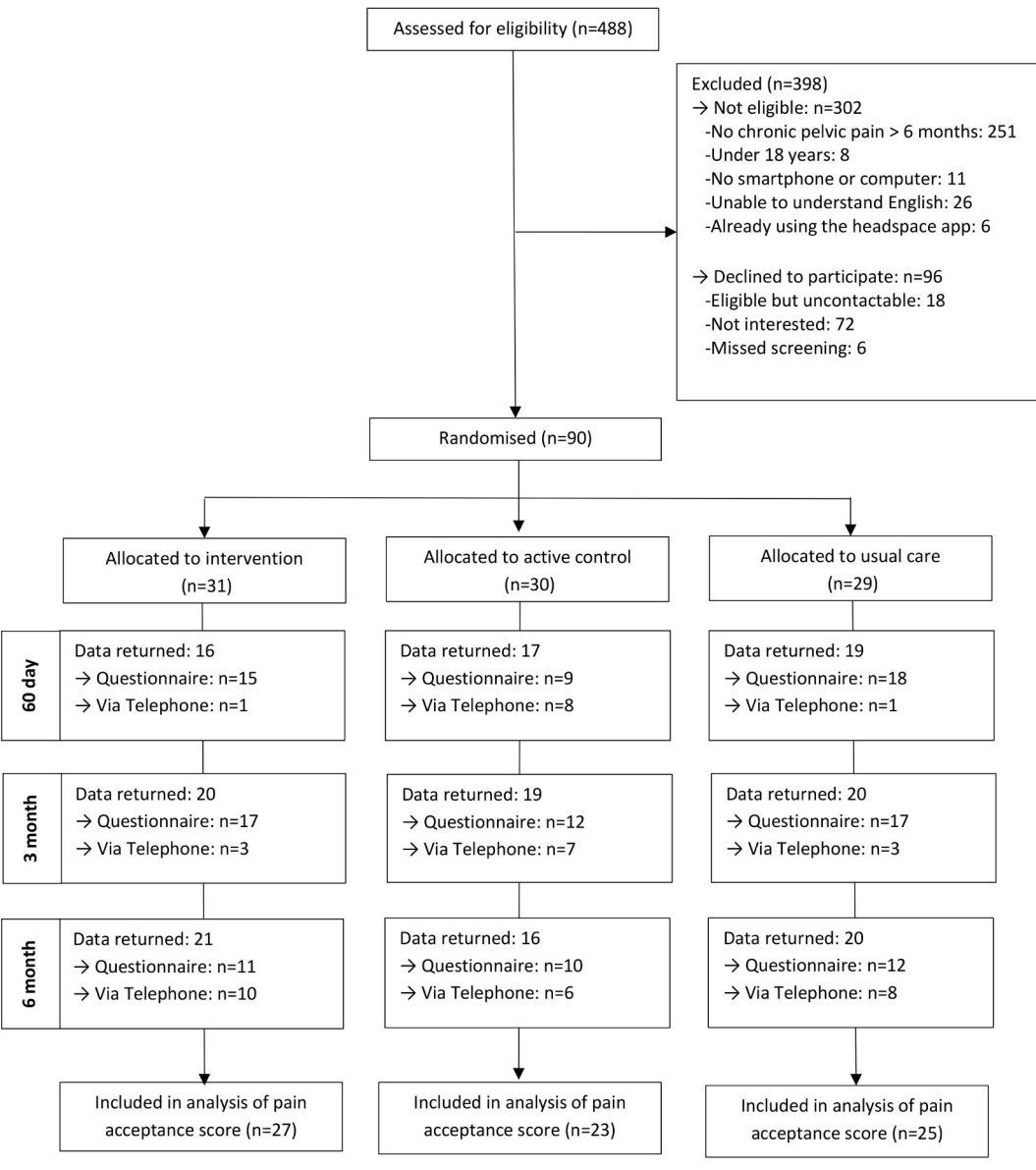

**Figure 1** Consolidated Standards of Reporting Trials diagram.

**Table 1** Baseline demographics and medical history

| | Summary measure | | |
| --- | --- | --- | --- |
| | Intervention (n=31) | Active control (n=30) | Usual care (n=29) |
| **Demographics** | | | |
| Age (years) | 34.8 (9.9) | 35.7 (5.7) | 35.0 (8.6) |
| Body mass index (kg/m$^2$) | 28.7 (7.0) | 26.2 (5.5) | 26.6 (6.3) |
| Living arrangements—n (%) | | | |
| Alone | 1 (3.3) | 2 (7.4) | 3 (11.1) |
| With others | 29 (96.7) | 25 (92.6) | 24 (88.9) |
| Employment status—n (%) | | | |
| Employed | 19 (63.3) | 18 (66.7) | 19 (67.9) |
| Unemployed and looking for work | 2 (6.7) | 0 (0.0) | 1 (3.6) |
| At school or in full time education | 2 (6.7) | 1 (3.7) | 4 (14.3) |
| Unable to work due to long term sickness | 4 (13.3) | 5 (18.5) | 1 (3.6) |
| Looking after your home/family | 3 (10.0) | 3 (11.1) | 2 (7.1) |
| Retired from paid work | 0 (0.0) | 0 (0.0) | 1 (3.6) |
| Age left full time education—n (%) | | | |
| Age 12 or less | 1 (3.3) | 1 (3.8) | 1 (3.6) |
| Age 13–16 | 9 (30.0) | 4 (15.4) | 3 (10.7) |
| Age 17–19 | 6 (20.0) | 5 (19.2) | 3 (10.7) |
| Age 20 or over | 11 (36.7) | 15 (57.7) | 16 (57.1) |
| Still in education | 3 (10.0) | 1 (3.8) | 5 (17.9) |
| Ethnic group—n (%) | | | |
| White | 10 (35.7) | 10 (43.5) | 15 (53.6) |
| Black | 6 (21.4) | 4 (17.4) | 3 (10.7) |
| Central Asian | 1 (3.6) | 1 (4.3) | 0 (0.0) |
| Middle Eastern | 0 (0.0) | 0 (0.0) | 1 (3.6) |
| Southern Asian | 8 (28.6) | 7 (30.4) | 3 (10.7) |
| Mixed | 0 (0.0) | 0 (0.0) | 2 (7.1) |
| Other ethnic group | 2 (7.1) | 1 (4.3) | 3 (10.7) |
| Do not wish to say | 1 (3.6) | 0 (0.0) | 1 (3.6) |
| Smoker—n (%) | | | |
| Yes | 8 (27.6) | 3 (12.5) | 6 (21.4) |
| No | 21 (72.4) | 21 (87.5) | 22 (78.6) |
| If yes, number of cigarettes per week | 23.9 (20.3) | 40.0 (20.0) | 47.6 (35.6) |
| Drink alcohol—n (%) | | | |
| Yes | 10 (34.5) | 9 (36.0) | 15 (55.6) |
| No | 19 (65.5) | 16 (64.0) | 12 (44.4) |
| If yes, number of units per week | 5.7 (5.3) | 8.3 (4.7) | 7.7 (7.2) |
| **Baseline medical history** | | | |
| Duration of pain—n (%) | | | |
| 0–6 months | 2 (6.7) | 0 (0.0) | 0 (0.0) |
| 7–12 months | 2 (6.7) | 4 (14.8) | 2 (7.1) |
| 1–2 years | 3 (10.0) | 5 (18.5) | 5 (17.9) |
| 3–5 years | 13 (43.3) | 7 (25.9) | 6 (21.4) |
| 6–10 years | 4 (13.3) | 4 (14.8) | 3 (10.7) |
| More than 10 years | 6 (20.0) | 7 (25.9) | 12 (42.9) |
| Pain over the past week (scale of 0–10) | 6.9 (2.3) | 5.8 (2.8) | 6.8 (2.3) |

Figures are mean (SD) unless stated otherwise.

**Table 2** App use. Figures are mean (SD) unless stated otherwise

| | Intervention (n=31) | Active control (n=28)* |
|---|---|---|
| Number of days a patient has used the app (within 60 days of randomisation) | 1.8 (4.3) | 7.0 (10.5) |
| Number of weeks a patient has used the app on 3 or more days (within the first 8 weeks from randomisation) | 0.3 (0.8) | 1.0 (1.6) |
| Used the app on 22 or more days within the first 60 days from randomisation—n (%) | 0 (0.0) | 2 (7.1) |
| Used the app on 3 or more days in 6 or more weeks (within the first 8 weeks from randomisation)—n (%) | 0 (0.0) | 0 (0.0) |
| Used the app on 22 or more days within the first 60 days and used the app on 3 or more days in 6 or more weeks within the first 8 weeks from randomisation—n (%) | 0 (0.0) | 0 (0.0) |

*Two participants in the active control group withdrew permission for their data to be used and are excluded from this analysis.

appendix 3, tables 1 and 2. Follow-up at 6 months was 68% in the mindfulness meditation group, 53% in the active control group and 69% in the usual care group. Follow-up rates by method of follow-up (phone or questionnaire), at different time points, and a comparison of baseline characteristics by questionnaire completion are given in online supplementary appendix 3, tables 3–5 and figure 1. The SD for CPAQ can be found in online supplementary appendix 3, table 6. Unintentional unblinding of treatment for either participants or researchers collecting data was rare (see online supplementary appendix 3, table 7).

App use was low in both groups, but was higher in the active control group than the intervention group (app used on mean 1.8 days intervention vs 7.0 active control— table 2). Few women used the app on more than 22 days within 60 days of randomisation (0 intervention vs 2 active control). Adherence to the app intervention was low or entirely absent across all other measures of app use (table 2). Daily app use within 60 days of randomisation is summarised in figure 2. The results from the app

usability questionnaire are shown in online supplementary appendix 3, tables 8 and 9.

### Clinical outcomes

We included 27 (87%) women from the intervention group, 23 (77%) from the active control group and 25 (86%) from the usual care group in the analysis of pain acceptance score. The 95% CIs for CPAQ (figure 3) rule out any strong benefit of the intervention compared with either the active control group or usual care group at any time point (higher CPAQ corresponds to better outcomes). The results for other clinical outcomes are consistent with no effect of the intervention (full results of clinical outcomes are shown in online supplementary appendix 3, tables 10–13 and figure 2).

### DISCUSSION

This trial shows that it is feasible to recruit women to a trial of a mindfulness meditation app. Follow-up rates were adequate and including data across all time points

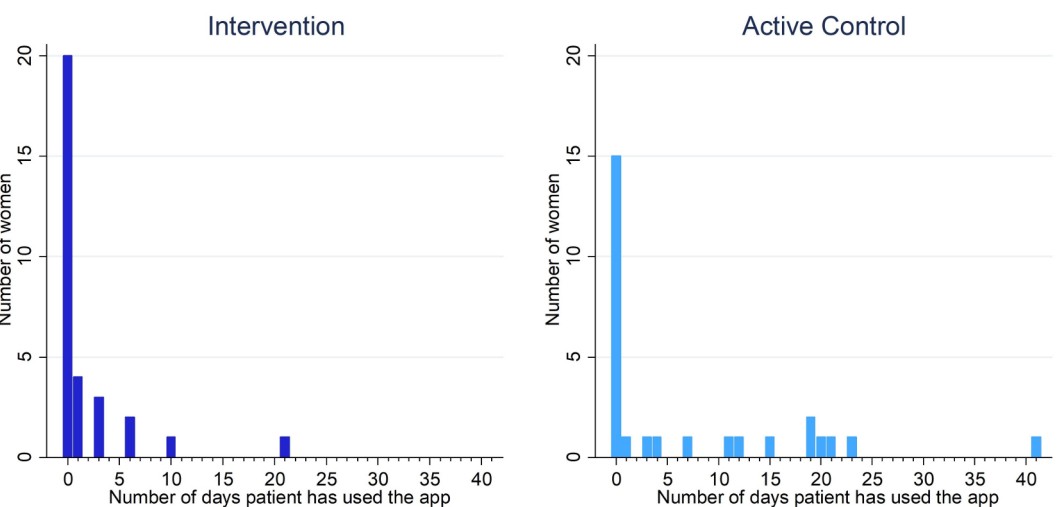

**Figure 2** Daily app use (defined as completing ≥90% of a session) within 60 days of randomisation in the intervention and active control groups.

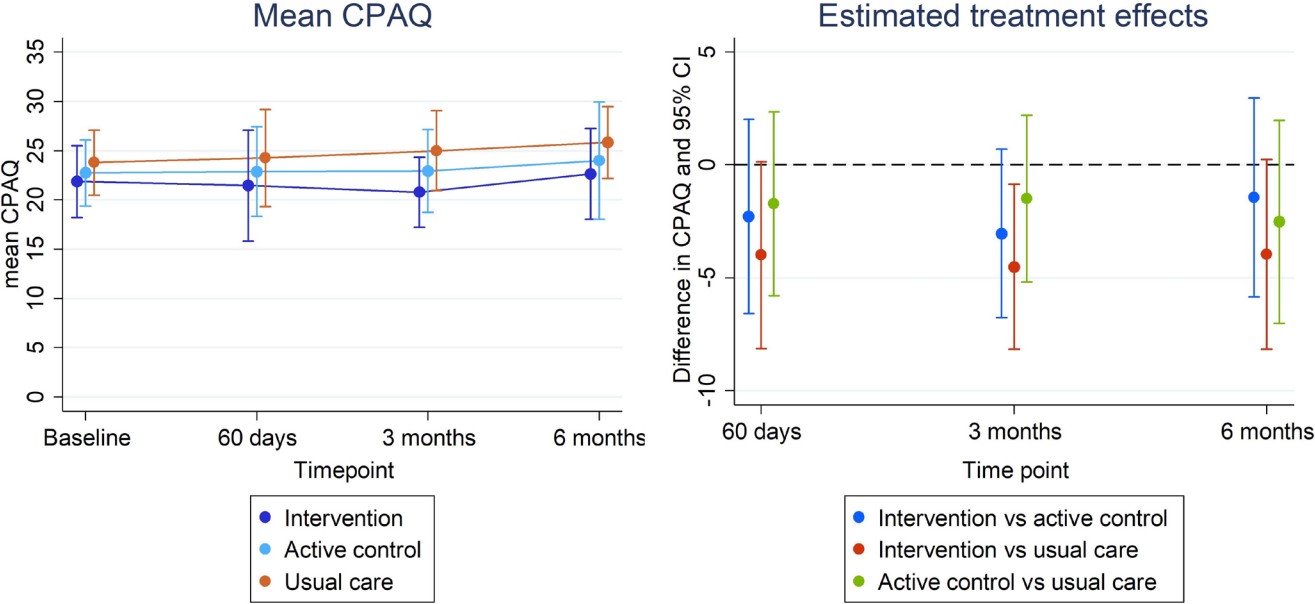

**Figure 3** Mean (95% CI) chronic pain acceptance score (CPAQ) and estimated treatment effect (95% CI) at each follow-up time point (CPAQ). Higher scores indicate better health outcomes.

meant that a relatively a high proportion of participants could be included in the analysis. This study provides estimates to inform sample size calculations for future research.

Most participants either did not complete any sessions on the apps or used them extremely infrequently. The analyses of clinical outcomes are consistent with no differences in health outcome between the three study arms. For pain acceptance, which was considered to be a likely outcome for a future effectiveness trial, our results suggest a meaningful effect of the mindfulness meditation app, delivered as it is in this trial, is unlikely. An effective intervention requires both engagement from those receiving it and the ability to change the targeted clinical outcome.[32] As engagement with the mindfulness meditation app evaluated in this study was very low it is unlikely it would be an effective intervention in the routine clinical setting for women with chronic pelvic pain, unless delivered as part of an intervention which significantly enhanced rates of engagement.

In addition to the work described in this paper we carried out in-depth qualitative interviews in order to examine the reasons for low levels of user engagement. Suggestions are given for improving the intervention such as co-development, an approach to intervention that involves the users in the design of the intervention. The findings are published in the companion paper describing the qualitative arm of this study.[15] The length of the intervention in this study (60 days) may also have been a barrier to participation and future work may want to explore different treatment lengths for remote-based mindfulness interventions.

An important lesson from this trial for future researchers was that intermediate follow-up points allowed for more participants to be included in the analysis of clinical outcomes than were followed up at the final time point. This demonstrates that utilising intermediate follow-up time points may help to minimise potential bias from missing data in trials.

The strengths of this study include randomisation of participants, which eliminates bias inherent in other designs such as before–after studies. We also blinded patients, recruiters and data collectors to which app group patients were allocated to. We used system-generated app data and therefore were able to obtain complete adherence data for all participants. One drawback to this method of data collection was that sessions of the app were only recorded as being complete if a participant listened to 90% of the session. This means this study may have underestimated app use if participants were only partially completing sessions. The levels of app use were so low, however, that this is unlikely to have had a material impact on the study's results. A second limitation is that recruitment was limited to two hospitals in one area of London, this may limit the generalisability of the results to settings where there is higher engagement with smartphone apps.

In conclusion, this study had high recruitment and adequate follow-up rates, demonstrating that it is feasible to conduct randomised trials in this patient population. However, due to extremely low adherence, further randomised trials to evaluate the benefit of the Headspace mindfulness meditation app for women with chronic pelvic pain are not warranted, unless additional

steps to improve engagement with the app are included in the intervention. Further discussion of reasons for low engagement and what could be done to improve engagement may be found in the qualitative part of this study.[15]

**Author affiliations**
[1]IoPPN, King's College London, London, UK
[2]Centre for Primary Care and Population Health, Queen Mary University of London, London, UK
[3]Department of Obstetrics and Gynaecology, Barts Health NHS Trust, London, UK
[4]Pelvic Pain Support Network, Poole, UK
[5]Women's Health Research Unit, Barts and The London School of Medicine and Dentistry, London, UK
[6]Faculty of Health Sciences, University of Southampton, Southampton, UK
[7]Department of Public Health, University of Granada, Granada, Spain
[8]Wolfson Institute of Preventive Medicine, Centre for Psychiatry, Queen Mary University of London, London, UK
[9]Centre for Primary Care and Public Health, Queen Mary University of London, London, UK
[10]Pragmatic Clinical Trials Unit, Queen Mary University of London, London, UK

**Contributors** EB conceived the research and lead the study. GF conducted the statistical analysis under the supervision of BCK. GF drafted the manuscript. GF, SN, CCC, JB, JD, LS, CR, KK, FR, ST, BCK and EB contributed to the design and conducted the study, and discussed and reviewed the final manuscript.

**Funding** This research was supported by the UK National Institute of Health Research, Research for Patient Benefit programme (RfPB PB-PG-1013-32025).

**Competing interests** None declared.

**Patient consent for publication** Not required.

**Ethics approval** Ethics approval was granted by Camden and Kings Cross Research Ethics Committee on 1 February 2016.

**Provenance and peer review** Not commissioned; externally peer reviewed.

**Data availability statement** Data are available upon reasonable request. Anonymised participant data are available upon reasonable request. Please contact pctu-data-sharing@qmul.ac.uk with any data sharing requests.

**ORCID iD**
Carol Rivas http://orcid.org/0000-0002-0316-8090

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
