## [Reviewer comments · BMJ Open]

ARTICLE DETAILS

TITLE (PROVISIONAL)	A smartphone app using psychological approaches for women with chronic pelvic pain presenting to gynaecology clinics (MEMPHIS): a randomised feasibility trial
AUTHORS	Forbes, Gordon; Newton, Sian; Cantalapiedra Calvete, Clara; Birch, Judy; Dodds, Julie; Steed, Liz; Rivas, Carol; Khan, Khalid; Röhrich, Frank; Taylor, Stephanie; Kahan, Brennan; Ball, Elizabeth

VERSION 1 - REVIEW

REVIEWER	Alexandra Crosswell UCSF, USA
REVIEW RETURNED	09-May-2019

GENERAL COMMENTS	This is a randomized trial of the mindfulness meditation app Headspace for a group of patients with chronic pelvic pain. This is definitely an area worthy of investigation given previous reports of mindfulness training helping patients with chronic pain. This main issue with this paper is that the methodology limits the ability to actually test the research question. First, by having a three arm intervention (Headspace, wait-list control, usual care), power was significantly reduced. It is unclear why both a wait-list control and usual care condition was needed. A greater number of participants in the active condition may have really assisted in the ability to detect an effect. However, the primary issue was with the methodology. There was not enough engagement with the participants to ensure that they actually engaged in the active intervention (Headspace). Participants thus did not actively participate in the intervention, which limits any interpretation that can be made to the intervention --- since it wasn't actually done! The authors conclude: "due to extremely low adherence, further randomised trials to evaluate the benefit of the Headspace mindfulness meditation app for women with chronic pelvic pain are not warranted at this time." -- however I disagree. With more appropriate engagement from research staff, participants may engage like they have in other studies of Headspace, and the results may be quite impactful. Finally, it is usual to have 'length of recruitment' as an outcome.
---

REVIEWER	Sarah Martin UCLA, USA
REVIEW RETURNED	24-May-2019

GENERAL COMMENTS	In the manuscript, “A smartphone app using psychological approaches for women with chronic pelvic pain (MEMPHIS): a randomised feasibility trial,” the authors describe the feasibility and usage of an adapted mindfulness meditation app in adults with chronic pelvic pain. The overall objective of this study addresses a significant clinical need; however, the presentation of the study aims, methods and results could be greatly improved. Primary Comments/Concerns:  1. The primary outcome(s) of the current study is (are) not clear. The study is initially presented as a feasibility study, but there are many outcomes listed in the methods section that are not introduced in the introduction or aims. The reference to pain acceptance as a “likely” primary outcome is also a concern. Was change in pain acceptance an aim for this study? If so, this needs to be specified in the introduction and aims, and justification for including this outcome also needs to be included. Further, a considerable part of the analyses section is devoted to assessing change in clinical outcomes and this is not listed as an aim. 2. More information pertaining to the study protocol and procedures are needed throughout. I understand that the full protocol is listed in the appendix, but it is unclear why this information is not summarized in the text of the manuscript. Most of my comments below address information that may be provided in the full protocol, but I think it is appropriate to provide some details in the manuscript to improve clarity for readers. I will defer to the editor and other reviewers regarding this issue. 3. There is a considerable amount of information provided in the appendices and limited data presented in the actual manuscript. It is unclear why the entire protocol, analyses plan, all data, and all tables are listed in appendices. I recommend adding some specific details to the manuscript and including descriptive tables only in the manuscript. 4. Collapsing the outcome data across all follow-up time points is a concern as data on changes in outcomes across time points are not provided. Introduction:  1. The introduction could be improved by focusing on the use of mindfulness in chronic pain and chronic pelvic pain specifically. For example, the mention of mindfulness in somatization syndromes is beyond the scope of the current paper. 2. I recommend rephrasing the statement on page 4, lines 17-19, as it is unclear what is meant by “activating the psychological trait or state of mindfulness.” 3. In the 4th paragraph of the introduction, the authors state that no studies have evaluated mindfulness via an app. Given that there have been studies examining app-based mindfulness interventions in other populations, I am wondering if the authors meant that there have not been studies in chronic pelvic pain. Please clarify. Methods  1. More information pertaining to the study protocol and procedures are needed throughout the methods section. I understand that the full protocol is listed in the appendix, but it is
---

unclear why this information is not summarized in the text of the manuscript. Most of my comments below address information that may be provided in the full protocol, but I think it is appropriate to provide some details in the manuscript to improve clarity for readers. I will defer to the editor and other reviewers regarding this issue.

2. Please define “organic” and “non-organic” chronic pelvic pain. Please consider using ICD diagnoses labels.

3. Please provide brief descriptions of the different intervention arms, including length of sessions and session frequency. Was usual care the third arm?

4. Further justification for the use of the Headspace app is needed. Please consider revising the phrase “quicker and cheaper” and proving some scientific justification.

5. Please expand on the 10-day mindfulness training required to access the app. Did all participants in the intervention arm complete this training in addition to the 60 day course?

6. A description of the informed consent procures is needed. In addition, how was data de-identified and transferred between the investigators and Headspace?

7. The outcomes section could be greatly improved. First, more justification for using each measure and criteria is needed. For example, the use of CPAQ standard deviation needs clarification and additional citations for its use parameter estimate. Further, how were the adherence criteria determined? Please consider writing out the clinical outcomes section and formatting this section appropriately. The clinical outcome measures listed address factors that go beyond the scope of the current aims and assessing these measures are not supported in the introduction. Please consider limiting the number of clinical outcomes for this feasibility study or providing appropriate justification for assessing these clinical outcomes.

8. The statement at the end of the outcomes section may be better suited in the introduction. Further, justification for the focus on chronic pain acceptance should be provided in the introduction and appropriate citations for this outcome in this population are needed.

9. How was the sample size number determined? Was a power analysis conducted?

10. The description of the PPI may be better suited in the description of the intervention as it relates to intervention development.

Results

11. Descriptive outcomes listed in table one should be listed in the methods section.

12. Given that recruitment was of interest in this study, more discussion or presentation of data from the CONSORT figure may be valuable. In addition, was the number of women recruited per recruitment period acceptable? How did the authors qualify this rate?

13. Was baseline medical history associated with any outcomes (feasibility or clinical outcomes)?

Discussion

14. It is stated that the results show that it was feasible to recruit women to this trial; however, an a priori cut-off criteria was not provided. How was this feasibility determined?

15. The discussion could be greatly improved by more discussion of the current findings in the context of the aims and discussing the current findings in the context of other literature on treatment adherence. Do the authors care to speculate why

	engagement may have been low in the current study? In its current form, the discussion seems to suggest that it is the mindfulness intervention in not an effective clinical intervention. Do the current data fully support this suggestion? Could there be other reasons for low engagement? Thank you for the opportunity to review this manuscript.
--	---

REVIEWER	Christina Bryant University of Melbourne Australia
REVIEW RETURNED	13-Jun-2019

GENERAL COMMENTS	Project title: A smartphone app using psychological approaches for women with chronic pelvic pain (MEMPHIS): a randomised feasibility trial This study has a number of significant strengths: it addresses an important clinical issue, has a well-articulated rationale, and the proposed methods are well-suited to address the key research questions. Although an apparent careful and relevant PPI process was undertaken, one of the key findings of the study is very important: there is a need to ensure that the PPI consultation process is truly reflective of the intervention's intended client groups. It was pleasing to see that feasibility criteria were well described, and that ease of recruitment emerged as a strength. More disappointing were the usage data – and the authors are open and honest in their consideration of where this shortcoming may have arisen. To this reviewer, the finding emphasises the need for a fully co-designed process that tailors the intervention to the intended client group. Tempting though it might be to use an existing app on the grounds of cost, it appears that this is not, in fact, viable. The authors are to be congratulated on their honest appraisal of the app's shortcomings.
---

REVIEWER	Julius Sim Keele University, UK
REVIEW RETURNED	07-Jul-2019

GENERAL COMMENTS	Title: 'randomised feasibility study' is certainly an accurate description, but I wonder if a slightly more specific term – and one that reflects the distinction that seems to be taking root between feasibility and pilot studies – might be 'external pilot'? P 2, line 27: strictly, '60 day' should be hyphenated (also P 5, line 23). P 4, line 37: needs an 'and' after 'resource-intensive,' and change 'require' to 'requires' and 'visit' to 'visits'. P 4, line 53: change 'estimates' to 'estimated'. P 5, line 30: insert 'the' before 'mindfulness group'.
--

	P 5, l 55: I'm not sure that 'strict allocation concealment' is very informative. I would either just say that there was allocation concealment, or indicate the specific ways in which it was 'strict'. P 6, line 48: change 'were' to 'was'. P 8, line 2: maybe 'precise' would be better than 'reliable', as the latter suggests the idea of reproducibility, which is not what you mean here, I think. The sample size is justified in terms of estimating the SD, but it might be worth commenting on its adequacy for estimates of binary outcomes, given that you assess these. From memory, I think the Teare et al paper that you cite comments on this. P 8, line 36: it is 'ethical approval' that is granted, not 'ethics'. Although consent is mentioned in the protocol, I don't think it is mentioned in the paper. P 9, line 11: I think this sentence needs to be more explicit. A CI 'excluding' a meaningful effect could mean one of two things. It could mean that the interpretation of the CI excludes the effect in the sense of making it implausible that it would be detected in a main trial, or it could mean that the CI lies above the meaningful effect, which indicates that the effect is in fact plausible in the main trial. Can you clarify your interpretation and how you reached it, as it is not clear from the text and Figure 3? Also, maybe say something about the findings vis-à-vis the clinical outcomes in the discussion, given that you have carefully estimated them with a statistical model and presented the findings.
--	--

VERSION 1 – AUTHOR RESPONSE

Reviewer(s)' Comments to Author:

Reviewer: 1

Reviewer Name: Alexandra Crosswell

Institution and Country: UCSF, USA

Please state any competing interests or state 'None declared': None declared

Please leave your comments for the authors below

We thank the reviewer for taking the time to review our article. Please see responses to the reviewer's comments below.

This is a randomized trial of the mindfulness meditation app Headspace for a group of patients with chronic pelvic pain. This is definitely an area worthy of investigation given previous reports of mindfulness training helping patients with chronic pain.

This main issue with this paper is that the methodology limits the ability to actually test the research question. First, by having a three arm intervention (Headspace, wait-list control, usual care), power was significantly reduced. It is unclear why both a wait-list control and usual care condition was needed. A greater number of participants in the active condition may have really assisted in the ability to detect an effect.

We agree with the reviewer that the sample size was not adequate to assess the effectiveness of the intervention. The reason for this was that MEMPHIS was a feasibility study whose primary objective was to assess whether it would be feasible to conduct a future definitive trial of the intervention. The design of this study, including sample size, was appropriate to address the question of feasibility.

We included both an active control app group and a usual care group to evaluate which would be better to use as the control arm in a future definitive trial; in particular, we wished to assess whether it was feasible to blind participants to the app by using an active control group. We note that we did not include wait list control in this trial.

We have edited the manuscript to make it clearer that this was a feasibility study where the main objective was to evaluate the feasibility of a future full scale trial.

However, the primary issue was with the methodology. There was not enough engagement with the participants to ensure that they actually engaged in the active intervention (Headspace). Participants thus did not actively participate in the intervention, which limits any interpretation that can be made to the intervention --- since it wasn't actually done!

The intervention considered in this study was to provide participants with the HeadSpace app with minimal additional support or encouragement to use it. We wished to evaluate the intervention as it would be used in practice, i.e. if it provided to participants as part of an UK national health service (NHS) appointment, but with minimal support outside of the initial appointment. This is reflective of how the app would be used within the scope of the NHS, which is the question most relevant to funding bodies and stakeholders within the NHS.

The lack of engagement of the intervention delivered with minimal support is an important finding of this study. We agree that the low levels of engagement limit the conclusions that can be drawn regarding the effectiveness of the headspace app if women were to engage with it. We have therefore edited our interpretations and conclusions to make it clear that the effectiveness of the headspace app may be different if delivered as part of an intervention which also improved engagement with the app from participants.

In addition, we have carried out in-depth qualitative interviews in order to examine the reasons for low levels of user engagement. The findings are published in the companion paper submitted to this journal.

The authors conclude: "due to extremely low adherence, further randomised trials to evaluate the benefit of the Headspace mindfulness meditation app for women with chronic pelvic pain are not warranted at this time." -- however I disagree. With more appropriate engagement from research staff, participants may engage like they have in other studies of Headspace, and the results may be quite impactful.

We have edited the discussion and conclusion to state that further research may be appropriate if additional steps to improve user engagement, such as co- development, an approach that involves the users in the design of the intervention, are added to the intervention.

Finally, it is usual to have 'length of recruitment' as an outcome.

We believe length of recruitment was an appropriate outcome for this study as whether recruitment is possible is a key area of uncertainty when planning a full randomised trial to evaluate the effectiveness of the intervention. For example, many publically funded trials fail to meet their sample size target (Carlisle, Kimmelman, Ramsay, & MacKinnon, 2015; Sully, Julious, & Nicholl, 2013); evidence that a trial would be able to recruit to target would be very important for funding bodies.

Reviewer: 2

Reviewer Name: Sarah Martin

We thank the reviewer for taking the time to review our article, in particular the detailed suggestions for improvements. Please see responses to the reviewer's comments below.

Institution and Country: UCLA, USA

Please state any competing interests or state 'None declared': none declared

Please leave your comments for the authors below Please see my attached comments

In the manuscript, "A smartphone app using psychological approaches for women with chronic pelvic pain (MEMPHIS): a randomised feasibility trial," the authors describe the feasibility and usage of an adapted mindfulness meditation app in adults with chronic pelvic pain. The overall objective of this study addresses a significant clinical need; however, the presentation of the study aims, methods and results could be greatly improved.

Primary Comments/Concerns:

1. The primary outcome(s) of the current study is (are) not clear. The study is initially presented as a feasibility study, but there are many outcomes listed in the methods section that are not introduced in the introduction or aims. The reference to pain acceptance as a "likely" primary outcome is also a concern.

We have reworded the objectives section of the study to make the primary objective more clear. Our primary objective was to evaluate the feasibility of implementing a randomised trial to assess the effectiveness of the intervention. We have also added to the introduction that measuring clinical outcomes was a secondary objective in this study.

As this study is a feasibility study we did not pre-specify a primary outcome. Pain acceptance was pre-specified in the protocol as a likely outcome for a future effectiveness trial however was not the primary outcome for this study.

Was change in pain acceptance an aim for this study? If so, this needs to be specified in the introduction and aims, and justification for including this outcome also needs to be included. Further, a considerable part of the analyses section is devoted to assessing change in clinical outcomes and this is not listed as an aim.

Assessing change in pain acceptance was not the primary aim of this study. We did however collect data on clinical outcomes as a secondary objective and analyse them using appropriate methods to 1) evaluate the feasibility of collecting this data in the population of interest. 2) Provide estimates and confidence intervals for the effect of the intervention.

We have added that measuring clinical outcomes, including pain acceptance, was a secondary objective to the introduction.

2. More information pertaining to the study protocol and procedures are needed throughout. I understand that the full protocol is listed in the appendix, but it is unclear why this information is not summarized in the text of the manuscript. Most of my comments below address information that may be provided in the full protocol, but I think it is appropriate to provide some details in the manuscript to improve clarity for readers. I will defer to the editor and other reviewers regarding this issue.

We agree with the reviewer that we may have not included sufficient detail of the methods in the main text of the manuscript. We have updated the manuscript to include additional details on the methods. Please see responses to specific comments below.

3. There is a considerable amount of information provided in the appendices and limited data presented in the actual manuscript. It is unclear why the entire protocol, analyses plan, all data, and all tables are listed in appendices. I recommend added some specific details to the manuscript and including descriptive tables only in the manuscript.

The entire protocol and analysis plan are included in the appendices as per journal requirements and to increase the transparency of reporting. There is not space in the manuscript to include all the content from these documents but the details contained in these documents may be of interest to some readers.

The results for clinical outcomes are given in the appendices of the manuscript as evaluating clinical outcomes was not the primary objective of this study. As the journal recommends 5 figures and tables in the manuscript we prioritised presenting data relating to the primary feasibility objectives of the study.

4. Collapsing the outcome data across all follow-up time points is a concern as data on changes in outcomes across time points are not provided.

In appendix 3, tables 12 and 13, we provide comparisons between arms for all outcomes at all time points. We do not report outcome data collapsed across all follow-up time points anywhere in the manuscript or appendices.

Introduction:

1. The introduction could be improved by focusing on the use of mindfulness in chronic pain and chronic pelvic pain specifically. For example, the mention of mindfulness in somatization syndromes is beyond the scope of the current paper.

We agree and have reworked the first three paragraphs of the introduction, introducing chronic pelvic pain, mindfulness meditation and its role in chronic pelvic pain.

2. I recommend rephrasing the statement on page 4, lines 17-19, as it is unclear what is meant by "activating the psychological trait or state of mindfulness."

We have edited the description of mindfulness meditation in the introduction and removed reference to "activating the psychological trait or state of mindfulness."

3. In the 4th paragraph of the introduction, the authors state that no studies have evaluated mindfulness via an app. Given that there have been studies examining app-based mindfulness interventions in other populations, I am wondering if the authors meant that there have not been studies in chronic pelvic pain. Please clarify.

We thank the reviewer for picking up this omission. We do mean there have been no studies in chronic pelvic pain and have edited the manuscript accordingly.

Methods

1. More information pertaining to the study protocol and procedures are need throughout the methods section. I understand that the full protocol is listed in the appendix, but it is unclear why this information is not summarized in the text of the manuscript. Most of my comments below address information that may be provided in the full protocol, but I think it is appropriate to provide some details in the manuscript to improve clarity for readers. I will defer to the editor and other reviewers regarding this issue.

We have added details to the manuscript in response to specific comments.

2. Please define “organic” and “non-organic” chronic pelvic pain. Please consider using ICD diagnoses labels.

We have reworded this to chronic pelvic pain with or without identifiable pathology’, and now introduce this concept in the background section.

3. Please provide brief descriptions of the different intervention arms, including length of sessions and session frequency. Was usual care the third arm?

We have added further details of the interventions to the manuscript including length of session and clarified that sessions were to be delivered daily. Usual care was the third arm and we have clarified this in the text.

4. Further justification for the use of the Headspace app is needed. Please consider revising the phrase “quicker and cheaper” and proving some scientific justification.

We have rewritten the background section making the scientific justification for considering mindfulness meditation for women with chronic pelvic pain more clear. We used an existing commercial app rather than designing one from scratch for practical reasons. We have rephrased to “We chose to evaluate an existing commercial app teaching mindfulness by guided meditation (Headspace Ltd) as it is approach seemed to save time and money compared to designing an app from scratch.”

5. Please expand on the 10-day mindfulness training required to access the app. Did all participants in the intervention arm complete this training in addition to the 60 day course?

We believe the reviewer is referring to the Take 10/foundation series referred to in the protocol. We have clarified in the manuscript that this is not separate training required prior to accessing the app but the first 10 days of mindfulness training received via the app. All participants in the intervention arm were offered this content as part of the intervention. As our results regarding adherence to the intervention show, few participants completed it.

6. A description of the informed consent procures is needed.

We have added a description of the informed consent procedure to the manuscript.

In addition, how was data de-identified and transferred between the investigators and Headspace?

We have added details to the manuscript on what data was transferred from headspace to the trial team. This was de-identified as included only access codes for the app and data on completed sessions. No data was transferred from the trial team to headspace.

7. The outcomes section could be greatly improved. First, more justification for using each measure and criteria is needed. For example, the use of CPAQ standard deviation needs clarification and additional citations for its use parameter estimate.

Standard deviation of CPAQ was collected as this is a key data point for a sample size calculation for a full trial. We have added this justification to the manuscript.

Further, how were the adherence criteria determined?

We determined definitions of adherence through discussions with the trial management group and trial steering committee. These definitions were chosen to give us a complete picture of how participants were using the app. We have added this justification to the manuscript.

Please consider writing out the clinical outcomes section and formatting this section appropriately.

The clinical outcome measures listed address factors that go beyond the scope of the current aims and assessing these measures are not supported in the introduction. Please consider limiting the

number of clinical outcomes for this feasibility study or providing appropriate justification for assessing these clinical outcomes.

The manuscript contains all clinical outcomes written in full, we are happy to reformat if the editors request this.

In response to earlier suggestion from this reviewer we have clarified our objectives to include measuring clinical outcomes. This makes it more clear that the clinical outcomes are within the scope of the objectives. We have included details of all prespecified clinical outcomes as this is best practice when reporting randomised feasibility trials (Eldridge et al., 2016).

8. The statement at the end of the outcomes section may be better suited in the introduction. Further, justification for the focus on chronic pain acceptance should be provided in the introduction and appropriate citations for this outcome in this population are needed.

We have moved this statement to the introduction.

We were considering pain acceptance for the follow up trial because it is a meaningful clinical outcome, that has been improved with mindfulness meditation in other pain conditions and is considered an important outcome for patients and clinicians. We have added this rationale and references to the manuscript to support its use.

9. How was the sample size number determined? Was a power analysis conducted?

No power analysis was conducted as the trial was not designed to evaluate the effectiveness of the intervention. The sample size was chosen to be sufficient to give a precise enough estimate of the standard deviation of pain acceptance for use in a sample size calculation for a full trial.

10. The description of the PPI may be better suited in the description of the intervention as it relates to intervention development.

We have included the description of the PPI in its own section as the PPI group contributed to study design and conduct as well as reviewing the intervention. In addition no changes to the intervention were made following this group so we do not see it as intervention development.

Results

11. Descriptive outcomes listed in table one should be listed in the methods section.

Table 1 is a baseline table of demographics measured prior to randomisation. As these are not post-randomisation measures we have not detailed them as outcomes in the methods section. This is in line with CONSORT reporting guidelines for clinical trials.

12. Given that recruitment was of interest in this study, more discussion or presentation of data from the CONSORT figure may be valuable. In addition, was the number of women recruited per recruitment period acceptable? How did the authors qualify this rate?

We did not report recruitment by recruitment period as this can be highly variable and depend on speed of site set up. The focus on the overall levels of recruitment was considered adequate to assess the feasibility of recruiting to a larger trial. Recruitment to target sample size within the period planned was considered a success. We are hesitant to deviate from our pre-specified feasibility measures by reporting outcomes in different ways.

13. Was baseline medical history associated with any outcomes (feasibility or clinical outcomes)?

We did not conduct this analysis as this was not one of our pre-specified objectives, and due to the small sample size in the trial it would not be possible to distinguish any differences observed from the play of chance.

Discussion

14. It is stated that the results show that it was feasible to recruit women to this trial; however, an a priori cut-off criteria was not provided. How was this feasibility determined?

Whilst no a priori cut off was given we considered the recruitment to be successful as the target sample size was achieved within the time period allocated to recruitment in the funding period.

15. The discussion could be greatly improved by more discussion of the current findings in the context of the aims and discussing the current findings in the context of other literature on treatment adherence. Do the authors care to speculate why engagement may have been low in the current study?

We have clarified the objectives of the study to make the emphasis on assessing the feasibility outcomes more clear. We believe that our discussion is in line with these aims. Further discussion on reasons for lack of engagement will be covered in a separate qualitative paper. We have added a sentence pointing the reader to this work to our discussion.

In its current form, the discussion seems to suggest that it is the mindfulness intervention in not an effective clinical intervention. Do the current data fully support this suggestion? Could there be other reasons for low engagement?

We have edited the discussion to clarify that we believe that the mindfulness intervention is not an effective intervention within this population, given the level of support offered. We believe that the data support this suggestion as without higher levels of engagement there is no way the intervention could be effective.

Thank you for the opportunity to review this manuscript.

Reviewer: 3

Reviewer Name: Christina Bryant

Institution and Country: University of Melbourne Australia

Please state any competing interests or state 'None declared': No COI

Please leave your comments for the authors below Project title: A smartphone app using psychological approaches for women with chronic pelvic pain (MEMPHIS): a randomised feasibility trial

This study has a number of significant strengths: it addresses an important clinical issue, has a well-articulated rationale, and the proposed methods are well-suited to address the key research questions. Although an apparent careful and relevant PPI process was undertaken, one of the key findings of the study is very important: there is a need to ensure that the PPI consultation process is truly reflective of the intervention's intended client groups.

It was pleasing to see that feasibility criteria were well described, and that ease of recruitment emerged as a strength. More disappointing were the usage data – and the authors are open and honest in their consideration of where this shortcoming may have arisen. To this reviewer, the finding emphasises the need for a fully co-designed process that tailors the intervention to the intended client group. Tempting though it might be to use an existing app on the grounds of cost, it appears that this

is not, in fact, viable. The authors are to be congratulated on their honest appraisal of the app's shortcomings.

We thank the reviewer for taking the time to review our manuscript. We agree with their assessment regarding the need for a fully co-designed process that tailors the intervention to the intended client group. This issue will be further discussed in a separate qualitative paper.

Reviewer: 4

Reviewer Name: Julius Sim

We thank the reviewer for reviewing our manuscript and for the helpful set of edits recommended.

Institution and Country: Keele University, UK

Please state any competing interests or state 'None declared': None declared

Please leave your comments for the authors below

Title: 'randomised feasibility study' is certainly an accurate description, but I wonder if a slightly more specific term – and one that reflects the distinction that seems to be taking root between feasibility and pilot studies – might be 'external pilot'?

We have included "feasibility" in the title of this article to be consistent with our previously published protocol. We are happy to change to "pilot" if the editor thinks this is appropriate.

P 2, line 27: strictly, '60 day' should be hyphenated (also P 5, line 23).

We have edited the manuscript.

P 4, line 37: needs an 'and' after 'resource-intensive,' and change 'require' to 'requires' and 'visit' to 'visits'.

We have edited the manuscript.

P 4, line 53: change 'estimates' to 'estimated'.

Edit made

P 5, line 30: insert 'the' before 'mindfulness group'.

Edit made

P 5, l 55: I'm not sure that 'strict allocation concealment' is very informative. I would either just say that there was allocation concealment, or indicate the specific ways in which it was 'strict'.

Edit amended to simply read "allocation concealment"

P 6, line 48: change 'were' to 'was'.

Edit made

P 8, line 2: maybe 'precise' would be better than 'reliable', as the latter suggests the idea of reproducibility, which is not what you mean here, I think.

Edit made

The sample size is justified in terms of estimating the SD, but it might be worth commenting on its adequacy for estimates of binary outcomes, given that you assess these. From memory, I think the Teare et al paper that you cite comments on this.

We have added a comment on the sample size being adequate for binary outcomes, referencing Teare et al.

P 8, line 36: it is 'ethical approval' that is granted, not 'ethics'. Although consent is mentioned in the protocol, I don't think it is mentioned in the paper.

Edit made, we have added details to the manuscript on the informed consent procedure.

P 9, line 11: I think this sentence needs to be more explicit. A CI 'excluding' a meaningful effect could mean one of two things. It could mean that the interpretation of the CI excludes the effect in the sense of making it implausible that it would be detected in a main trial, or it could mean that the CI lies above the meaningful effect, which indicates that the effect is in fact plausible in the main trial.

Can you clarify your interpretation and how you reached it, as it is not clear from the text and Figure 3?

We have edited the description of the confidence interval to say "rules out any strong benefit of the intervention" to make interpretation clearer for the reader.

Also, maybe say something about the findings vis-à-vis the clinical outcomes in the discussion, given that you have carefully estimated them with a statistical model and presented the findings.

We have added an additional sentence to the discussion vis-à-vis the clinical outcomes.

VERSION 2 – REVIEW

REVIEWER	Sarah Martin UCLA Department of Pediatrics, USA
REVIEW RETURNED	03-Oct-2019

GENERAL COMMENTS	Reviewer 2 R1 Comments: The resubmission of this manuscript improved significantly from the initial submission and it is evident that the authors took great care in addressing the reviewers' comments. The authors have adequately addressed most of my initial comments. My remaining comments are listed below: 1. Comment in response to original comment #1: I thank the authors for their response to this comment and appreciate the edits. I caution the authors on the use of the term "effectiveness" as this has certain implications in clinical trials etc. I recommend stating that this was a feasibility study and removing references to testing effectiveness. For example, the first sentence of the primary objective paragraph could be rephrased to: "The primary objective of the study was to assess the feasibility of implementing a randomized trial of a mindfulness meditation intervention delivered by a smartphone app for women with chronic pelvic pain." I also think that feasibility could be the primary outcome of this trial. Further, the examination of effects on clinical outcomes could be described as "exploratory" aims/objectives.a. Given the large number of clinical outcomes assessed, I recommend referencing the type of clinical outcomes in the objectives paragraph. For example, the authors could state something referencing that they examined a variety of outcomes that assessed quality of life, functioning etc.
---

	2. Comment in response to original intro comment #1: I appreciate the authors' edits to the mindfulness description in the background. I have a few remaining comments/suggestions. a. First, the transition to discussing mindfulness could be improved. One suggestion would be to combine the second and third paragraphs. Specifically, inserting the description of mindfulness after the first sentence of the third paragraph and finishing the paragraph with the limitations of the current literature. b. Second, I recommend describing the practice of mindfulness more generally. Not all mindful meditations involve focusing on the breath. You may want to consider using "the present moment" instead and maybe reference focusing on the breath as an example. 3. Please provide a citation after the first sentence of the fourth paragraph (line 14). Citing MBSR treatment may be helpful. 4. Comment in response to original comment #15: In the introduction, the authors imply that the length of in-person mindfulness interventions may be a barrier to intervention participation. In the discussion, it may be warranted to mention the length of the mindfulness app intervention and whether future studies should examine the treatment length of remote-based mindfulness interventions. Minor Comments: - On page 153, line 19, please consider changing "provided" to "received" Thank you for the opportunity to review this manuscript.
--	---

REVIEWER	Julius Sim Keele University, UK
REVIEW RETURNED	30-Sep-2019

GENERAL COMMENTS	Thank you for responding to my comments. Small point, but in the explanation added of the confidence intervals on page 10, you need 'rule', not 'rules' - something for the proof stage!
--

VERSION 2 – AUTHOR RESPONSE

Reviewer(s)' Comments to Author:

Reviewer: 4

Reviewer Name: Julius Sim

Institution and Country: Keele University, UK

Please state any competing interests or state 'None declared': None declared

Please leave your comments for the authors below Thank you for responding to my comments.

Small point, but in the explanation added of the confidence intervals on page 10, you need 'rule', not 'rules' - something for the proof stage!

Edit made, many thanks.

Reviewer: 2

Reviewer Name: Sarah Martin

Institution and Country: UCLA Department of Pediatrics, USA

Please state any competing interests or state 'None declared': None declared

Please leave your comments for the authors below Reviewer 2 R1 Comments:

The resubmission of this manuscript improved significantly from the initial submission and it is evident that the authors took great care in addressing the reviewers' comments. The authors have adequately addressed most of my initial comments. My remaining comments are listed below:

1. Comment in response to original comment #1: I thank the authors for their response to this comment and appreciate the edits. I caution the authors on the use of the term "effectiveness" as this has certain implications in clinical trials etc. I recommend stating that this was a feasibility study and removing references to testing effectiveness. For example, the first sentence of the primary objective paragraph could be rephrased to: "The primary objective of the study was to assess the feasibility of implementing a randomized trial of a mindfulness meditation intervention delivered by a smartphone app for women with chronic pelvic pain." I also think that feasibility could be the primary outcome of this trial. Further, the examination of effects on clinical outcomes could be described as "exploratory" aims/objectives.

We have edited the primary objective paragraph removing the word effectiveness from the first sentence.

We would prefer to describe the examination of clinical effects simply as secondary objectives rather than exploratory. As the outcomes and their analysis were pre-specified we do not think exploratory is the best way to describe them as exploratory is frequently used to describe post hoc, not pre-specified outcomes or analysis. We are happy to take editorial guidance on this point.

We have not stated that feasibility is the primary outcome for this study, as feasibility encompasses several different outcomes rather than a single outcome, none of which can be described as primary. Furthermore, this was not pre-specified as the primary outcome in the trial protocol, and we feel it would be inappropriate to change this at this stage. We therefore believe it is sufficient for feasibility to be described as the primary objective.

a. Given the large number of clinical outcomes assessed, I recommend referencing the type of clinical outcomes in the objectives paragraph. For example, the authors could state something referencing that they examined a variety of outcomes that assessed quality of life, functioning etc. We have added a sentence describing the type of outcomes assessed.

2. Comment in response to original intro comment #1: I appreciate the authors' edits to the mindfulness description in the background. I have a few remaining comments/suggestions.

a. First, the transition to discussing mindfulness could be improved. One suggestion would be to combine the second and third paragraphs. Specifically, inserting the description of mindfulness after the first sentence of the third paragraph and finishing the paragraph with the limitations of the current literature.

We have merged the second and third paragraphs as suggested

b. Second, I recommend describing the practice of mindfulness more generally. Not all mindful meditations involve focusing on the breath. You may want to consider using “the present moment” instead and maybe reference focusing on the breath as an example.
We have changed the description of mindfulness meditation to refer to the present moment, and mentioned breath as an example.

3. Please provide a citation after the first sentence of the fourth paragraph (line 14). Citing MBSR treatment may be helpful.
We have added a reference to “Mindfulness meditation in clinical practice”, Paul Salmon et al, 2004 which describes MBSR treatment.

4. Comment in response to original comment #15: In the introduction, the authors imply that the length of in-person mindfulness interventions may be a barrier to intervention participation. In the discussion, it may be warranted to mention the length of the mindfulness app intervention and whether future studies should examine the treatment length of remote-based mindfulness interventions.
We’ve added a sentence to the discussion mentioning the length of the app intervention:

“The length of the intervention in this study (60 days) may also have been a barrier to participation and future work may want to explore different treatment lengths for remote based mindfulness interventions.”

Minor Comments:

- On page 153, line 19, please consider changing “provided” to “received”
Edit made

Thank you for the opportunity to review this manuscript.
Many thanks for the detailed and helpful suggestions which have much improved this paper.

VERSION 3 - REVIEW

REVIEWER	Sarah Martin UCLA, USA
REVIEW RETURNED	13-Dec-2019

GENERAL COMMENTS	The resubmission of the revised manuscript improved significantly. The authors have adequately addressed all of my comments and I have no further concerns or suggestions. Thank you for the opportunity to review this manuscript.
--